# Plastid ancestors lacked a complete Entner-Doudoroff pathway, limiting plants to glycolysis and the pentose phosphate pathway

Sonia E. Evans [1], Anya E. Franks [1], Matthew E. Bergman [1], Nasha S. Sethna [1], Mark A. Currie[1,2] & Michael A. Phillips [1,2] ✉

The Entner–Doudoroff (ED) pathway provides an alternative to glycolysis. It converts 6-phosphogluconate (6-PG) to glyceraldehyde-3-phosphate and pyruvate in two steps consisting of a dehydratase (EDD) and an aldolase (EDA). Here, we investigate its distribution and significance in higher plants and determine the ED pathway is restricted to prokaryotes due to the absence of *EDD* genes in eukaryotes. EDDs share a common origin with dihydroxy-acid dehydratases (DHADs) of the branched chain amino acid pathway (BCAA). Each dehydratase features strict substrate specificity. *E. coli* EDD dehydrates 6-PG to 2-keto-3-deoxy-6-phosphogluconate, while DHAD only dehydrates substrates from the BCAA pathway. Structural modeling identifies two divergent domains which account for their non-overlapping substrate affinities. Coupled enzyme assays confirm only EDD participates in the ED pathway. Plastid ancestors lacked EDD but transferred metabolically promiscuous EDA, which explains the absence of the ED pathway from the Viridiplantae and sporadic persistence of *EDA* genes across the plant kingdom.

Multiple catabolic pathways exist in living cells for extracting energy from glucose. Although most organisms rely principally on the Embden–Meyerhof–Parnas (EMP) pathway (i.e., glycolysis) and the pentose phosphate pathway (PPP) for bioenergetic metabolism, alternative routes such as the Entner–Doudoroff pathway (ED pathway)[1] degrade glucose to pyruvate with fewer enzymes, a property favored by some single-celled prokaryotes living under conditions of nitrogen limitation[2]. The ED pathway acts as a shunt which converts PPP-derived 6-phosphogluconate (6-PG) to pyruvate and glyceraldehyde-3-phosphate (GAP) in two enzymatic steps (Fig. 1a). This shunt begins with the dehydration of 6-PG to 2-keto-3-deoxy-6-phosphogluconate (KDPG) catalyzed by 6-PG dehydratase (EDD). KDPG is then cleaved to pyruvate and GAP by KDPG aldolase (EDA). Glycolysis and its major shunts have different ATP/protein efficiencies[3], and the optimal route for a given organism may reflect its environmental conditions, such as carbon and nitrogen availability. The ED pathway yields half as much ATP per mole of glucose as glycolysis. Organisms which can produce ATP through photosynthesis or aerobic respiration are less subject to this constraint and may favor the ED pathway. Indeed, aerobic bacteria capable of photosynthesis or oxidative PPP tend to rely more heavily on the ED pathway than glycolysis, whereas strict and facultative anaerobes use glycolysis almost exclusively; a limited number of prokaryotes use both[4]. In aquatic, photomixotrophic environments where the Calvin-Benson-Bassham cycle (CBC) runs in parallel to glycolysis and the PPP in cyanobacteria, the ED pathway may play roles in maintaining CBC intermediates through anaplerotic reactions[5].

[1]Department of Cell and Systems Biology, University of Toronto, Toronto, ON M5S 3G5, Canada. [2]Department of Biology, University of Toronto—Mississauga, Mississauga, ON L5L 1C6, Canada. ✉e-mail: michaelandrew.phillips@utoronto.ca

The committing step of the ED pathway is the EDD-catalyzed dehydration of 6-PG to KDPG, the only metabolic intermediate unique to this pathway. EDDs (EC 4.2.1.12) belong to the dehydratase subfamily known as hydro-lyases which includes dihydroxy acid dehydratases (DHADs; EC 4.2.1.9). The latter dehydrate 2,3-dihydroxymethylvalerate (DMV) and 2,3-dihydroxyisovalerate (DIV) to 2-ketomethylvalerate (KMV) and 2-ketoisovalerate (KIV), respectively, in the branched chain amino acid (BCAA) pathway. Both catalyze the dehydration of vicinal diol acids to an enol intermediate, followed by tautomerization to their corresponding 2-keto acids (Fig. 2a, b)[6–8], and both are oxygen-sensitive iron-sulfur proteins[7,9]. This stands in contrast to the mechanism of unrelated dehydratases such as the short-chain dehydrogenase/reductase superfamily (SDR). For instance, dTDP-D-glucose 4,6-dehydratase (EC 4.2.1.46)

**Fig. 1 | The Entner–Doudoroff pathway and sequence analysis of EDD and DHAD. a** Schematic of the main routes of glucose catabolism. The Entner–Doudoroff (ED) pathway is shown in blue. **b** Selected domains from a multiple sequence alignment of representative dihydroxy acid dehydratases (DHAD) and 6-phospho-gluconate (6-PG) dehydratase (EDD) amino acid sequences highlighting the defining motifs of each class. Residues in dark and light gray represent 80% and 60% similarity cutoffs, respectively. The yellow box signifies domains unique to EDDs while the blue box is unique to DHADs. Species names are as follows: Eco *Escherichia coli*, Sen *Salmonella enterica*, Aba *Acinetobacter baumannii*, Tma *Thermotoga maritima*, Sel *Synechococcus elongatus*, Nos *Nostoc* *sp.*, Ath *Arabidopsis thaliana*, Gma *Glycine max*, Smo *Synechococcus moorigangaii*, War *Candidatus Woesearchaeota archaeon*, Zmo *Zymomonas mobilis*, Atu *Agrobacterium tumefaciens*. The full sequence alignment can be found in Supplementary Fig. 2. Asterisks indicate residues conserved among EDD involved in 6-PG substrate binding. **c** Phylogenetic analysis of DHAD and EDD proteins generated using maximum likelihood method from MEGA X with bootstraps values shown as a percentage from 1000 replicates. The outer, curved orange and blue lines display EDD and DHAD sequences, respectively. Protein accession numbers are listed in Supplementary Table 1.

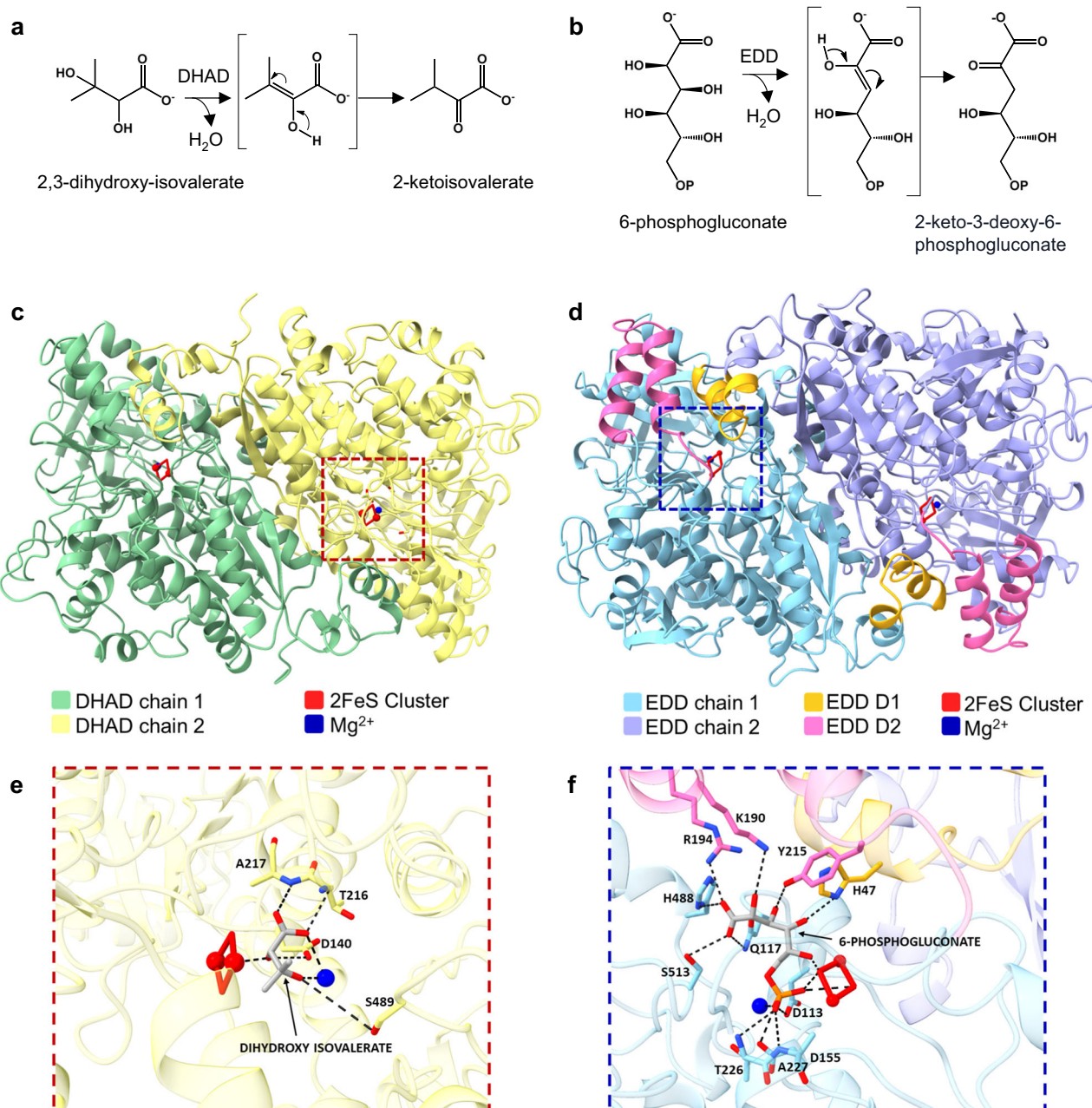

**Fig. 2 | Structural models of dihydroxy acid dehydratase (DHAD) and 6-phosphogluconate dehydratase (EDD). a** Dehydration of 2,3-dihydroxyisovalerate by DHAD in the branched chain amino acid pathway. **b** Dehydration of 6-phosphogluconate by EDD in the ED pathway. OP signifies a phosphate group. **c** The crystal structure of the *Arabidopsis thaliana* DHAD (PDB ID: 5ZE4) depicting a crystallographic dimer in green and yellow and (**d**) our model of the *Escherichia coli* EDD (ACA77431) dimer in cyan and purple with domains 1 (D1) and 2 (D2) shown in orange and pink, respectively. **e** Predicted substrate binding of 2,3-dihydroxyisovalerate in the active site of DHAD. **f** The analogous binding of 6-phosphogluconate in the EDD active site. The reciprocal docking experiments did not support binding when the substrates were switched. The magnesium ion and 2Fe-2S clusters of each protein are shown in blue and red, respectively. Dotted black lines in (**e**) and (**f**) denote predicted polar contacts between substrate and surrounding residues and ions.

relies on NAD$^+$ as a cofactor and first oxidizes the substrate to a ketone intermediate, followed by dehydration and reduction of the resulting double bond[10]. The shared mechanisms of EDD and DHAD and high degree of amino acid similarity suggest they are paralogs derived from a common evolutionary precursor. The potential of EDDs and DHADs to recognize multiple substrates has not been evaluated.

Due to their common evolutionary origin and high degree of amino acid similarity, DHADs and EDDs have been treated collectively in phylogenetic analyses aimed at understanding the natural distribution of the ED pathway[11]. The assumption that they have overlapping substrate specificities and biochemical functions[12] has led to the conclusion that a functional ED pathway operates in flowering plants, mosses, ferns[11], and diatoms[13] in addition to prokaryotes[4]. Bacteria possess both types of dehydratase, but in plants a definitive physiological role has thus far only been established for DHAD[14].

The potential existence of the ED pathway in plant cells would have major implications for our understanding of the regulation of central carbon metabolism. For instance, in the cyanobacterium *Synechocystis sp*. PCC 6803, the ED pathway is thought to play a role in anaplerosis to replenish the CBC[5]. The impact of a functional ED pathway in multi-compartmental plant cells would be unclear. Parallel versions of glycolysis and the PPP function in chloroplasts and the cytosol, but expression of plastidic isoforms catalyzing irreversible steps is generally timed to avoid futile cycling with the CBC[15]. Diversion of hexose phosphate through an ED pathway in the cytosol would therefore have different implications than a similar shunt in the chloroplast. A transcript identified as an *EDD* in the diatom *Phaeodactylum tricornutum* encodes a protein predicted to localize to the mitochondria based on bioinformatic analysis[13], but this has not yet been confirmed by direct localization or functional characterization

The existence of a functional ED pathway in plants would have major implications for engineering native metabolic pathways in the chloroplast that depend on a supply of pyruvate and GAP, such as the 2-*C*-methyl-D-erythritol-4-phosphate (MEP) pathway[16]. Indeed, upregulation of ED pathway genes in *E. coli* increased flux through the MEP pathway[17]. The effects of the ED pathway in plants would depend heavily on subcellular localization.

Although a characterization of plant EDA proteins has conclusively demonstrated their activity in vitro[11], the functional characterization of an EDD from a eukaryotic source has not yet been reported. Thus, the distribution of the ED pathway in the Viridiplantae (and other eukaryotes) is currently uncertain. Here we show that the ED pathway does not naturally occur in plants or in any biological lineage outside of prokaryotes. Furthermore, we show that its absence from the green eukaryotic lineages is likely a consequence of its absence from the ancestral cyanobacterial endosymbionts which evolved into plastids during early eukaryote evolution. In contrast, *EDA* genes, which were uniformly present in the presumed last common ancestors of plastids, persist sporadically across the plant kingdom, likely due to their ability to complement other aldolase reactions of central metabolism through their broad substrate specificity.

## Results

### 6-phosphogluconate dehydratase genes are absent from eukaryotic genomes

An amino acid alignment of plant and microbial dehydratase sequences from the BCAA pathway (DHADs; EC 4.2.1.9) and the ED pathway (6-PG dehydratases or EDDs; EC 4.2.1.12) identified several mutually exclusive, conserved motifs that were specific to each dehydratase group (Fig. 1b and Supplementary Fig. 1). The most prominent were two domains in the N-terminal half which differed significantly between these two groups (Fig. 1b). In the first domain, sequences annotated as bacterial EDDs encoded a conserved 16-17 residue motif corresponding to amino acids 42–57 in the *E. coli* protein and containing the consensus sequence

LAHGFAAX$_4$(D/E)KX$_3$ (Fig. 1b, domain 1). This motif is absent in the corresponding DHAD sequences from those same bacterial species and in representative DHAD sequences of plants. A second region (Fig. 1b, domain 2), corresponding to amino acids 185–218 in *E. coli*, is enriched in basic residues and is highly conserved among EDDs. It includes the consensus motif KXK(V/I)RQLYAXXK. The corresponding region in DHADs, corresponding to residues 152–192 of the *E. coli* sequence, is 6–7 residues longer, rich in hydrophobic residues, and highly variable with no conserved consensus motif. In addition to these insertions and deletions, our comparison also identified more than a dozen short (2–4 residues) motifs absolutely conserved in EDDs with highly variable sequences in the corresponding positions of DHAD proteins (Supplementary Fig. 1). None of the representative plant dehydratase sequences encoded the domains associated with the group that included confirmed bacterial EDDs. These diagnostic motifs suggested that EDDs and DHADs could be differentiated by their primary sequences, a hypothesis we tested further below.

Using these diagnostic motifs as predictors of function, a phylogenetic analysis of 155 EDD and DHAD protein sequences with high similarity to *E. coli* EDD suggested that *EDD* is restricted to prokaryotes (Fig. 1c and Supplementary Table 1). EDD homologs were common in archaea and proteobacteria, infrequent in cyanobacteria, purple bacteria and rhizobia, and overrepresented in aerobes, consistent with the distribution reported by Flamholz et al.[4]. EDD amino acid sequences formed a distinct clade separate from DHADs, suggesting early divergence of *EDD* genes from an ancestral *DHAD* sequence, possibly through a gene duplication. DHADs, in contrast, are broadly distributed across prokaryotes, fungi, and streptophytes (Fig. 1c), and our DHAD single gene phylogeny matched the broad features of their species phylogenies, including the late-branching Zygnematophyceae, recently identified as the descendant of the most recent common ancestor between land plants and algae[18–20]. Our search ultimately failed to identify EDD homologs in genomes of animals, protists, fungi, or any member of the Viridiplantae (Supplementary Table 1). Several apparent exceptions to this rule were observed in draft genome or transcript sequences of plants, fungi, and animals (Supplementary Data 1). However, an analysis of their codon usage, phylogenetic grouping, and the observed lack of transit peptides and introns suggest they are bacterial sequencing artefacts (Supplementary Data 1). Furthermore, the absence of similar EDD homologs in basal angiosperms, fungi, or animals makes the re-emergence of an *EDD* gene in these species unlikely. The absence of *EDD* genes from genomes of eukaryotes suggests that the ED pathway does not play a role in glucose metabolism outside of prokaryotic lineages.

About a quarter of heterotrophic bacteria use the ED pathway[4], but its distribution in autotrophs such as cyanobacteria has not been well defined. We considered whether absence of *EDD* genes from eukaryotes was better explained by gene loss from prokaryotic genomes during endosymbiosis or mere absence in the plastid ancestor. A review of sequence databases failed to identify *EDD* genes among the likely cyanobacterial descendants of plastid ancestors. Although the true sister group to plastids is uncertain, recently proposed descendants sharing the most recent common ancestor to plastids and free-living cyanobacteria coalesce around *Gleomargarita lithospora* and several closely related taxa which include *Pseudanabaena sp*., *Synechococcus sp*., *Synechocystis sp*., *Prochlorococcus marinus*, *Trichodesmium sp*., *Oscillatoria sp*., and *Arthrospira sp*.[21–23]. Previous analyses viewed nitrogen-fixing members of the Nostocales and Stigonematales as the closest descendants[24], and early or late divergence of plastids within cyanobacterial phylogeny remains a contentious issue[20,25]. Currently available sequence data indicate that few members of these groups encode an *EDD* gene, while all possess *DHAD* and *EDA* (Supplementary Table 2). We identified only four examples of EDD homologs among all cyanobacteria: *Synechococcus moorigangaii*, *Nostoc sp*. 3335mG, and *Leptolyngbya sp*. 15MV, and *Leptolyngbya valderiana* BDU 20041

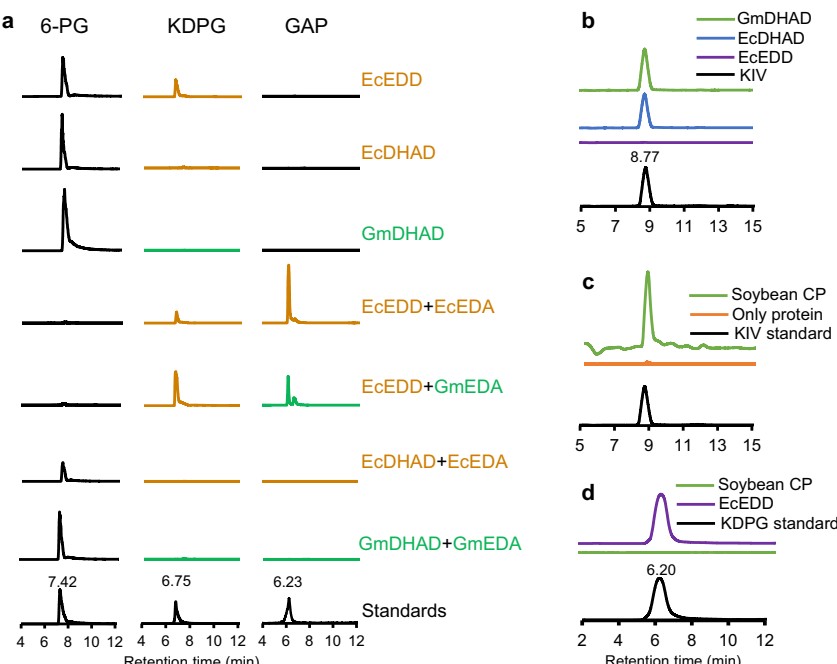

**Fig. 3 | Enzyme assays of 6-phosphogluconate dehydratase (EDD) and dihydroxyacid dehydratase (DHAD).** Soybean (Gm) DHAD and *E. coli* (Ec) EDD and DHAD were expressed in *E. coli*, and purified proteins were used individually or in combination to evaluate respective activity of EDD and DHAD. **a** Coupled ED assays using 6-phosphogluconate as the substrate demonstrate that EDD, but not DHAD, provides KDPG which can be further converted to GAP and pyruvate by KDPG aldolase (EDA) of plant or bacterial origin. Bacterial proteins and product chromatograms are shown in brown; plants in green. **b** DHAD assay using 2,3-dihydroxyisovalerate (DIV) to produce 2-ketoisovalerate (KIV). Both *E. coli* (Ec) and soybean DHAD converted DIV to KIV, while *E. coli* EDD did not. **c** DHAD assay using crude protein (CP) extract from soybean and DIV supplied as substrate displaying detectable DHAD activity in soybean leaf extracts. **d** The same soybean crude protein extract supplied with 6-phosphogluconate did not yield the expected EDD dehydration product, 2-keto-3-deoxy-6-phosphogluconate (KDPG). Each reaction was repeated at least once with identical results.

(Supplementary Table 2). EDD genes are notably absent from the modern descendants of cyanobacteria proposed to have diverged just before or after plastids (*G. lithospora*, *Pseudanabaena* PCC6802 and PCC7367, and *Synechococcus sp.* JA-2-3B a 2-13). These rare examples of cyanobacterial EDD sequences in the genomic record do not form a single clade but rather are interspersed among proteobacterial and archaeal sequences (Fig. 1c). When overlaid onto recent cyanobacterial species phylogenies[22,23], their distribution is suggestive of acquisition through lateral gene transfer (LGT) (Supplementary Fig. 2).

## Dihydroxy acid dehydratases and 6-phosphogluconate dehydratases are structurally similar but differ in substrate binding motifs

To understand how differences in EDD and DHAD primary sequence impact protein structure and function, we next carried out protein modeling and substrate docking experiments with representative structures of each dehydratase class. We used the crystal structure of the *A. thaliana* DHAD (PDB ID: 5ZE4) (Fig. 2c) and an AlphaFold[26] model of the *E. coli* EDD (Fig. 2d) as our representative structures to compare DHAD and EDD active site architectures (Fig. 2e, f). Both of these enzymes have highly similar α/β folds that pack into a single globular structure. The domain 1 consensus sequence that we observed exclusively in EDD enzymes encompasses α3, the first four amino acids of α4, and the connecting α3–α4 loop in *E. coli* EDD (Fig. 2d). Together with α2, domain 1 forms a three-helix bundle that sits at one end of the globular EDD fold. The second EDD domain consensus sequence we identified is located at the distal end of the molecule and forms a lid over the catalytic center of the enzyme, which includes a 2Fe-2S cluster and magnesium ion (Fig. 2f). Domain 2 is comprised of α9, α10, the α9–α10 loop, the last three amino acids of the β4-α9 loop, and the first five amino acids of the α10–α11 loop.

Both EDD and DHAD dehydratases form dimers in solution that interact in a head-to-tail fashion, which places domain 1 from one EDD protomer next to the catalytic site and domain 2 of the second protomer within the dimer. Domain 1 from protomer 1 contacts the β1-α5 loop, α6, α6–α7 loop, and domain 2 of protomer 2. The conserved histidine of the domain 1 consensus sequence along with the conserved lysine and arginine from the EDD domain 2 consensus sequence point into the substrate binding pocket. The equivalent side chains in the *A. thaliana* DHAD are all hydrophobic (Fig. 2e). Therefore, EDD domains 1 and 2, and the equivalent structural elements in DHAD enzymes form a large portion of the catalytic pocket and contribute significantly to the selection of substrate size, shape, and electrostatic characteristics.

To evaluate how these changes might impact substrate specificity in our representative EDD and DHAD enzymes, we performed in silico docking experiments with both substrates. In our substrate docked model of *E. coli* EDD, the conserved residues H47, K190, and R194 from domain 1 and 2 consensus sequences make polar contacts with the carboxyl and hydroxyl moieties of our docked 6-PG (Fig. 2f). All of these bonds are absent in DHAD enzymes. However, our docking experiments with *A. thaliana* DHAD indicate that it can accommodate its preferred substrate, 2,3-dihydroxyisovalerate (DIV), which is much smaller than 6-PG and has two hydrophobic methyl groups (Fig. 2e). When the substrates were switched, 6-PG docking into DHAD fails to meet the AutoDock significance threshold (−0.2), and DIV is positioned at a distance that is too far to interact with the catalytic site of EDD. Together, these observations suggest differences in domains 1 and 2 between EDD and DHAD account for their preferred substrate specificities, and that overlap in substrate recognition is unlikely based on our substrate docking experiments. We next proceeded to test this notion with

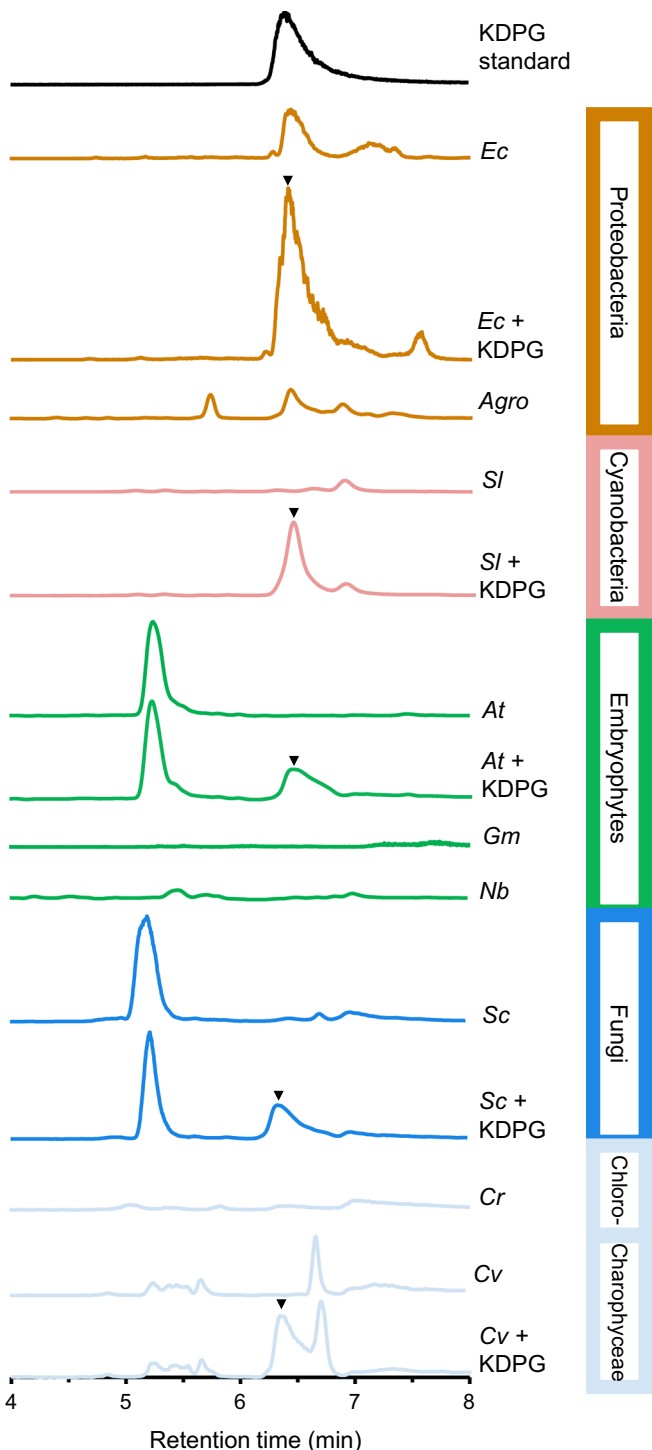

**Fig. 4 | Detection of 2-keto-3-deoxy-6-phosphogluconate (KDPG) in cells of representative prokaryotes and eukaryotes.** Liquid chromatography–tandem mass spectrometry multiple reaction monitoring of KDPG was performed in negative mode, and the vertical axis represents the detector response for its characteristic transition ($m/z$ 257 → 79). All graphs are plotted on the same absolute scale. Colors indicate phylogenetic group as per Fig. 1 (and in vertical boxes). KDPG standard (black), bacteria (brown), Embryophytes (green), fungi (blue), cyanobacteria (salmon), and algae (light blue). Tissue samples spiked with KDPG are marked with a triangle where the added KDPG appears in the chromatogram. Ec *Escherichia coli*, Agro *Agrobacterium tumefaciens*, Sl *Synechococcus leopoliensis*, At *Arabidopsis thaliana*, Gm *Glycine max*, Nb *Nicotiana benthamiana*, Sc *Saccharomyces cerevisiae*, Cr *Chlamydomonas reinhardtii*, Cv *Chara vulgaris*. The gray vertical bar shows the expected retention time of KDPG. Each extraction was repeated at least once with identical results.

biochemical characterization of members of both groups using DIV and 6-PG as substrates.

## 6-phosphogluconate dehydratase and dihydroxy acid dehydratase genes encode high fidelity enzymes with no overlapping substrate recognition

Purified, recombinant *E. coli* DHAD (AAA67574), *E. coli* EDD (ACX39449), and *G. max* DHAD (KAH1217623) (Supplementary Fig. 3) were assayed using a substrate from the BCAA pathway (DIV) or the ED pathway (6-PG). *E. coli* EDD catalyzed the expected dehydration of 6-PG to KDPG (Fig. 3a) but had no detectable reactivity toward DIV under similar reaction conditions (Fig. 3b), even after 24 h of incubation. Conversely, the soybean and *E. coli* DHAD both converted DIV to its corresponding dehydration product 2-ketoisovalerate (KIV) but displayed no detectable product formation when supplied with 6-PG for 24 h (Fig. 3a, b). A total protein extract of soybean leaves readily converted DIV to KIV (Fig. 3c) but failed to dehydrate 6-PG to KDPG under similar reaction conditions (Fig. 3c). These observations demonstrate that DHADs and EDDs catalyze similar dehydrations but have no overlap between their respective substrates. It further suggests that in soybean leaves, flux through the ED pathway is insignificant.

We next conducted coupled enzyme assays of the full ED pathway sequence (Fig. 3a) to screen the ability of various combinations of purified EDD, EDA, and DHAD proteins to support flux through an in vitro ED pathway. When *E. coli* EDD and EDA were supplied with 6-PG, the 6-PG was completely consumed while GAP accumulated as the end product with a low level of KDPG still detectable (Fig. 3a), confirming that these in vitro assays could monitor the complete catalytic conversion of 6-PG to GAP by EDD and EDA. Complete conversion to GAP and pyruvate was also observed when purified EDA from soybean was used. However, when the *E. coli* EDD was exchanged for the *E. coli* DHAD or the DHAD from soybean, formation of KDPG was not detected (Fig. 3a). Together, these results demonstrate DHADs cannot replace genuine EDDs in the two step ED pathway metabolic sequence from 6-PG to GAP and pyruvate.

## Prokaryotic but not eukaryotic cells contain 2-keto-3-deoxy-6-phosphogluconate, the metabolite unique to the Entner–Doudoroff pathway

To further test the notion that the presence or absence of a functional EDD sequence in the genome was a predictor of a functional ED pathway, we searched for KDPG in metabolite extracts of representative prokaryotic and eukaryotic organisms. We hypothesized that KDPG, the metabolic intermediate unique to the ED pathway, would be detected in prokaryotic cells containing a genuine EDD in their genome but not in eukaryotes. Liquid chromatography - mass spectrometry (LCMS/MS) analysis of KDPG in polar metabolite extracts of proteobacteria (*E. coli* and *A. tumefaciens*), whose genomes encode both EDD and EDA, demonstrated the presence of KDPG (Fig. 4), while KDPG was absent from extracts prepared from eukaryotic sources including *S. cerevisiae* (fungi), *C. reinhardtii* (Chlorophyceae), *C. vulgaris* (Charophyceae), or the leaves or roots from various embryophytes (soybean, tobacco, and Arabidopsis). Extracts of *Synechococcus leopoliensis* cells, whose genome lacks an EDD, similarly contained no KDPG (Fig. 4). Standard addition with authentic KDPG confirmed the observed peak in prokaryotic samples to be KDPG and indicated concentrations in *E. coli* and *A. tumefaciens* of $4.7 \pm 0.82$ and $2.8 \pm 0.75$ µmol g$^{-1}$ cells fresh weight, respectively. An isobaric background peak in the *C. vulgaris* extract was similarly ruled out as KDPG by the same method, and here addition of KDPG led to two separate peaks (Fig. 4). We calculated limits of detection and quantification of 2.5 and 8.3 pmol per assay, respectively. Complete details on the chromatographic separation and analytical detection of KDPG are provided in Supplementary Fig. 4. In

summary, KDPG could be detected in prokaryotic cells whose genome included a genuine *EDD* gene[4] but not in eukaryotic cells, consistent with our phylogenetic (Fig. 1) and structural modeling analysis (Fig. 2). These metabolite profiling data support the notion that the presence of genuine *EDD* and *EDA* genes in the genome is an effective predictor for the presence of a functional ED pathway in an organism.

## Plant 2-keto-3-deoxy-6-phosphogluconate aldolases (EDAs) catalyze other aldol reactions of central metabolism

EDA catalyzes the aldol cleavage of KPDG into pyruvate and GAP. EDA genes are widely distributed in plants and microbes[11]. We incubated soybean aldolase (GmEDA) with three aldolase substrates to better understand the persistence of this gene in plant genomes despite the absence of an *EDD* homolog. Purified, recombinant GmEDA (Supplementary Fig. 3) truncated to remove the predicted transit peptide converted KDPG into GAP and pyruvate in equimolar amounts, as confirmed by LCMS/MS and GCMS, respectively (Fig. 5a, b), and this activity disappeared when substrate or protein was withheld or when the protein was boiled prior to initiating the assay. The pseudomature protein catalyzed the aldol cleavage of KDPG into pyruvate and GAP with a $K_M$ of 310 μM (Fig. 5c), consistent with previous literature reports for this reaction[27].

EDAs show broad specificity for substrates of other aldolase reactions[28,29], a property consistent with the substrate promiscuity of aldolases generally[30]. When fructose-1,6-bisphosphate was used as a substrate, GmEDA readily carried out the analogous cleavage to yield GAP and dihydroxyacetone phosphate (DHAP), albeit with a substantially higher $K_M$ of 1.26 mM (Fig. 5d). When incubated with erythrose-4-phosphate and DHAP, the enzyme produced sedoheptulose-1,7-bisphosphate (SBP) through the corresponding reverse reaction (Supplementary Fig. 5). This substrate promiscuity confirmed that plant EDAs could participate in multiple aldolase reactions and carry out moonlighting roles elsewhere in plant metabolism.

The subcellular localization of GmEDA in the chloroplast was confirmed by agroinfiltration of *N. benthamiana* (Fig. 5e). When expressed as a full-length pre-protein with eGFP fused to its C-terminus, GmEDA produced punctate foci of fluorescence that overlapped with chlorophyll autofluorescence (Fig. 5f), similar to that observed for the chloroplast localized PII protein[31], Arabidopsis 1-deoxy-D-xylulose-5-phosphate reductoisomerase[32] and tomato 1-deoxy-D-xylulose-5-phosphate synthase isoforms 1 and 2[33].

These results confirmed that plant EDAs are localized to the chloroplast where they can participate in diverse aldolase reactions of glycolysis and the CBC. Compared to *E. coli* EDD, which displays high substrate fidelity and only dehydrates 6-PG, the affinity of EDAs for additional substrates may serve to optimize carbon flux in central metabolism and thus be subject to positive selective forces which has resulted in their retention in plant genomes.

## Discussion

The glycolytic shunt initially described by Entner and Doudoroff more than 70 years ago[1] provides a rapid source of GAP and pyruvate with a minimal input of genetic and biochemical machinery. This discovery provided insight into a third major route of glucose oxidation in prokaryotes. Early metabolic studies in *Entamoeba histolytica*[34], *Aspergillus niger*[35], and *Penicillium notaturn*[36] suggested it was also a significant metabolic route in unicellular or filamentous eukaryotes (reviewed in[2]). More recently, the enzymatically similar DHAD class of dehydratases from the BCAA pathway has been implicated in the ED pathway, leading to speculation that the ED pathway operates in plants and diatoms[11–13]. Based on biochemical characterization of DHAD and EDD substrate specificity (Fig. 3), the structural analysis of class-specific, conserved motifs responsible for conferring this specificity (Fig. 2), evidence in the genomic record (Fig. 1), and direct detection of the ED

pathway-specific metabolite, KDPG, in prokaryotes but not eukaryotes (Fig. 4), we conclude that not only is the ED pathway absent from photosynthetic eukaryotes such as flowering plants, conifers, bryophytes, ferns, mosses, and algae, it is absent from eukaryotic lineages altogether. Although we cannot rule out the future discovery of an ED pathway-capable eukaryote adapted to an unusual environment, currently available evidence suggests that the ED pathway is restricted to prokaryotes.

We considered two explanations for why eukaryotes do not possess genes for the ED pathway. One possibility is that EDD genes were deleted from an ancestral endosymbiotic cyanobacterium that possessed a full ED pathway due to a metabolic or other incompatibility with a mitochondriate eukaryotic host. This explanation would be consistent with the observation that the Archaeplastida host which engulfed an ancestral cyanobacterial endosymbiont was likely a facultative anaerobe[37], a metabolic scenario which would disfavor the ED pathway in early eukaryotic cells[4]. Indeed, anaerobic metabolism is widespread among eukaryotes[38]. Deletion of *EDD* from the plastid ancestor would effectively block flux through the ED pathway, based on our observation that DHAD cannot substitute EDD activity. Indeed, deletion of prokaryotic genes while others were being transferred to the nuclear genome was common[39]. However, our analysis suggests that the ancestor of plastids simply lacked *EDD* genes (Supplementary Table 2). We found few examples of cyanobacteria encoding both *EDD* and *EDA*, implying the complete ED pathway is uncommon among autotrophic prokaryotes. This is consistent with previous reports that the majority of prokaryotes possessing the full complement of ED pathway genes are heterotrophic proteobacteria and actinobacteria[4].

Plastids are nested within cyanobacterial lineages, and ongoing efforts aim to identify their point of divergence from other cyanobacterial groups since primary endosymbiosis. Such insights would prove highly relevant to understanding the distribution of the ED pathway. A recent effort places the divergence point of plastid ancestors after *Synechococcus* sp. JA-2-3B a 2-13 but before *Pseudanabaena* (sp. PCC7367 and sp. PCC6820)[22] (Supplementary Fig. 2), which differed slightly from prior analyses that had placed the divergence of deeply branching *Pseudanabaena* before that of plastids[23,40]. These recent reports agree that orders such as Nostocales and Oscillatoriales diverged even later. Placement of plastids remains controversial with contemporaneous reports arguing for a classification of plastids as late-branching[25]. Future analyses will surely refine the topology of the cyanobacterial species tree further. These ongoing refinements are relevant to rationalizing the absence of the ED pathway among the eukaryotic green lineages. This situation is made more complex by frequent LGT among free living prokaryotes[41]. Whereas genes acquired via the plastid through endosymbiotic gene transfer (EGT) are sequestered in the eukaryotic lineage, their counterparts in cyanobacteria continue to undergo LGT and other forms of prokaryotic recombination[42]. If the true plastid ancestor contained a full complement of genes for the ED pathway but lost it early in endosymbiosis, we would expect that at least a few closely related cyanobacterial sister groups (*G. lithospora*, *Pseudanabaena sp.*, *Synechococcus sp.*) would have retained the EDD gene, along with at least some early diverging members of the Archaeplastida (Glaucophytes and Rhodophytes) if not directly in Charophyte or Chlorophyte lineages, where we observe no evidence of a functional ED pathway (Fig. 3). By comparison, we do observe broad distribution of DHAD genes in both plants and algae (Fig. 1e), including among Zygnematophyceae, the closest algal relatives of land plants[18,19,43]. It is well known that gene loss through deletion has potential to confound identification of true sister groups[42], but uniform loss of EDD genes among every member of the Archaeplastida following inheritance from a cyanobacterial symbiont is harder to rationalize.

In contrast, if the true plastid ancestor did not encode a homolog of EDD, then it never would have been transferred to plant nuclear

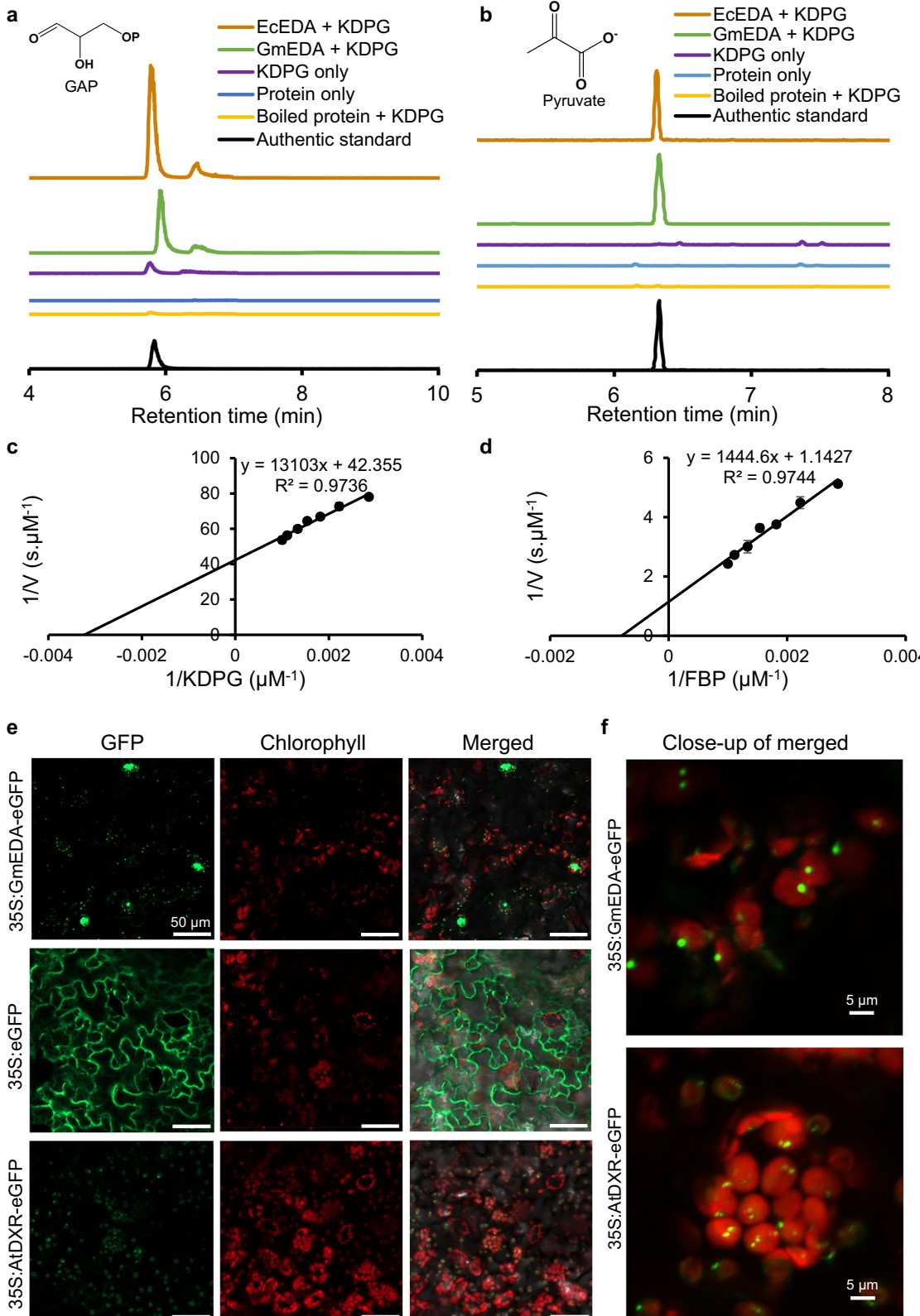

**Fig. 5 | Substrate promiscuity and subcellular localization of KDPG aldolase (EDA).** Chromatograms of reaction products GAP (**a**) and pyruvate (**b**) from an enzymatic assay of soybean and *E. coli* EDA when KDPG was supplied as substrate. GAP and pyruvate were analyzed by LCMS/MS and GC-MS, respectively. Authentic standards of GAP and pyruvate were used to confirm identities. Lineweaver-Burke plots for 2-keto-3-deoxy-6-phosphogluconate (KDPG) aldolase (EDA) activity are shown using KDPG (**c**) and D-fructose-1,6-bisphosphate (FBP) (**d**) as substrate at concentrations ranging from 0.25–1 mM (error bars signify standard deviation; *n* = 3). **e** Subcellular localization of GmEDA in agroinfiltrated *Nicotiana*

*benthamiana*. Confocal images show green eGFP fluorescence, chlorophyll auto-fluorescence, and their overlays indicating GmEDA-eGFP colocalization with chloroplasts as punctate foci. Free eGFP and *Arabidopsis thaliana* 1-deoxy-D-xylulose 5-phosphate reductoisomerase (AtDXR) act as positive controls for cytosol and chloroplast localization, respectively. Bar = 50 μm. **f** Close-up of merged images for GmEDA and AtDXR showing colocalization detail of the eGFP fusion proteins with chloroplasts (bar = 5 μm). For (**e**) and (**f**), the experiment was performed twice with identical results.

genomes through EGT and would be uniformly absent from plastid and nuclear genomes across the Archaeplastida. Our observations support this scenario. Moreover, we would still expect that over the next 1.5 billion years (approximately the time since the origin of plastids), some cyanobacteria would acquire EDD genes through LGT. But in that case, these cyanobacterial EDD homologs would not cluster together in single gene phylogenies in a way that is congruent with their accepted species phylogenies obtained from ribosomal RNA sequences[44]. This is precisely the distribution of EDD genes among cyanobacteria we observe (Fig. 1c and Supplementary Fig. 2), which branch more closely with proteobacterial and archaeal homologs rather than forming a single clade of cyanobacterial EDD sequences. Our position would become untenable by the future discovery of *EDD* and *EDA* gene pairs in a cyanobacterial species more closely related to modern day plastids than *G. lithospora*, particularly if such a hypothetical EDD gene nested monophyletically with other cyanobacterial EDD sequences. However, currently available evidence suggests that the absence of a complete ED pathway in the plastid ancestor provides the most parsimonious explanation for its absence among the Archaeplastida.

*Synechocystis* PCC 6803 has been used as a model to study ED pathway in cyanobacteria[5,11,45,46], but efforts to measure flux in the ED pathway in this strain were not successful[47]. The hyperthermophile Archaeon *Sulfolobus solfataricus* utilizes semi- and non-phosphorylated versions of the ED pathway featuring gluconate, 2-keto-3-deoxygluconate (KDG), and KDPG[28,48,49]. Chen et al. have speculated that *Synechocystis* DHAD (WP_010874288 or *slr0452*) might dehydrate gluconate to KDG, followed by phosphorylation to KDPG through the semi-phosphorylated pathway[11]. Our analysis indicates that *Synechocystis* PCC 6803, like the majority of cyanobacteria, does not employ a complete ED pathway due to the absence of a genuine *EDD* gene.

There is likely little evolutionary advantage to the Viridiplantae obtaining the genes to employ the ED pathway. While the ED pathway reportedly plays an anapleurotic role in prokaryotes[5], its potential effects in multi-compartmental eukaryotic cells are less predictable. In plant cells where glycolysis and the PPP operate in parallel in multiple compartments, the ED pathway could potentially exert a cataplerotic effect on the CBC by diverting 6-PG to pyruvate, depending on the compartment where these activities were localized. A functional ED pathway in the cytosol of plant cells would disrupt the glucose-6-phosphate shunt that replenishes the CBC in higher plants[50,51]. The lower yield of the ED pathway compared to glycolysis[4,52] would offer plants few advantages, and the benefit of reducing the cost of protein production may be insufficient to offset the loss of ATP yield when nitrogen is not limiting.

The persistence of EDA in plant genomes may be due to their substrate promiscuity which may serve auxiliary roles in carbon metabolism. Our results confirmed the catalytic activity of EDA toward substrates involved in two aldolase condensations in the chloroplast (Fig. 5d and Supplementary Fig. 5): the condensation of GAP and DHAP to form fructose-1,6-bisphosphate and the condensation of DHAP and erythrose-4-phosphate to form SBP. The former may additionally run in reverse as part of glycolysis and the PPP. A contributing role of EDA to these reactions may serve to optimize flux through these pathways of central metabolism. EDA-deficient mutants in *Synechocystis sp.* 6803 were reported to exhibit reduced fitness under photomixotrophic conditions[5,11]. Consistent with this, increased carbon flux through the CBC was observed in Δ*eda* mutants of *Synechocystis* under steady state conditions, but the same Δ*eda* deletion mutant displayed a growth defect under fluctuating light conditions, suggesting a potential role of EDA in carbon assimilation in cyanobacteria[46]. Given the growth phenotype of *Synechocystis* Δ*eda* mutants[5,45] together with the substrate promiscuity and multifunctional nature of EDA we report here, it is likely that EDA participates elsewhere in primary metabolism and might offer selective advantages to organisms, allowing them to adapt

to diverse environmental conditions and optimize carbon utilization strategies. However, focused follow-up studies are needed to probe the role of EDA in plant metabolism.

The absence of the ED pathway from higher plants makes the application of synthetic biology approaches to install this ancestral pathway an enticing possibility. The distribution of the ED pathway among cyanobacteria suggests its absence in the plastid ancestor best explains its absence among plants today, rather than evolutionary forces favoring its deletion. This implies it may be feasible to manipulate this alternative route in plants. This would serve to better understand the metabolic implications of installing the ED pathway in a multi-compartment cell. Additionally, synthetic biology approaches would probe the potential of this pathway to augment the production of high-value natural products. For instance, the products of the ED pathway, GAP and pyruvate, are the two substrates needed for the MEP pathway[16], which in turn supplies the synthesis of tens of thousands of diverse plastid-derived isoprenoids by generating the universal isoprenoid intermediates, isopentenyl diphosphate and dimethylallyl diphosphate. The MEP pathway is the engine for the biosynthesis of many medically useful natural products[53], and optimizing a restored ED pathway in plants to increase the availability of GAP and pyruvate for plastidic isoprenoid biosynthesis would offer many advantages over efforts to upregulate endogenous sources, including circumventing regulatory restrictions on these broadly utilized central metabolites. A recent report coupling an upregulated ED pathway in *E. coli* to production of isoprene indicated that the ED pathway and MEP pathway may be natural partners for synthetic biology approaches aimed at improving isoprenoid production in model systems[17]. Future efforts aimed at increasing provision of 6-PG to power a synthetic ED pathway will therefore explore the ability of synthetic biology to redirect central metabolism to support plant isoprenoid biosynthesis and other pyruvate-dependent pathways using this long-overlooked alternative to glycolysis.

## Methods

### Plant and bacterial cultivation
*Arabidopsis thaliana* ecotype Columbia 0 plants were grown in 3" pots in environmental chambers at 21 °C under short day conditions (8 h light/16 h dark) with 150 µE m$^{-2}$ s$^{-1}$ white light and 65 ± 10% relative humidity. *Nicotiana benthamiana* were grown in 4" pots in a long day chamber (16 h light/8 h darkness) at 25 °C with the same light intensity and humidity. *Glycine max* were grown in 4" pots under greenhouse conditions in pots with BX Promix soil. *Chara vulgaris* was grown in a freshwater aquarium with continuous filtration and aeration (12 h day length). *Synechococcus leopoliensis* (CPCC 102) was obtained from the Canadian Phycological Culture Centre and grown in BG-11 media with 18–20 µE m$^{-2}$ s$^{-1}$ white light. All plant or algal tissue was flash frozen in liquid nitrogen and ground to a fine powder prior to lyophilization to dryness against a vacuum of 25 µbar.

### Cloning, protein expression, and biochemical characterization
Genes encoding the *E. coli* EDD [ACX39449], EDA [WP_114414364] and DHAD [AAA67574] proteins, *G. max* (soybean) EDA[XP_014617853], and DHAD [KAH1217623] were PCR amplified using primers which included attB1 (forward) and attB2 (reverse) adapter sequences for cloning into the Gateway entry vector pDONR207 (Invitrogen) (see Supplementary Table 3 for all primer sequences) according to manufacturer's instructions. Pseudomature forms of plant genes without the transit peptide encoding sequence were amplified in parallel. Following BP clonase recombination and bacterial transformation, resistant colonies obtained on Luria-Bertani (LB)-plates containing gentamycin at 50 µg ml$^{-1}$ were screened by PCR for the expected insert size, and purified plasmids were obtained from overnight liquid cultures of the same media and antibiotic. Clones encoding the expected sequence were then subcloned into a Gateway-compatible pET28 vector[54] for

expression in *E. coli* BL21(DE3) cells. For transient expression in *N. benthamiana*, Gateway amplified genes were transferred into the pB7FWG2 vector[55] with LR clonase (Invitrogen). Unless otherwise noted, full-length genes including transit peptides were expressed in plants whereas predicted transit peptides were removed for expression in bacteria. Bacterial cultures (200 ml) were induced at the exponential growth stage with 1 mM isopropyl β-D-1-thiogalactopyranoside for 16 h at 16 °C. All protein extraction and purification steps were carried out at 4 °C. Cell pellets were harvested by centrifugation and resuspended in 1/10 vol protein extraction buffer. For aldolase expression, this consisted of 50 mM $KH_2PO_4$ at pH 8.0 with 300 mM NaCl while for dehydratase-expressing cultures, the buffer was 100 mM Tris-HCl at pH 8 with 100 mM KCl, 0.5% glycerol and 0.05% tween-20. Both buffers contained a plant protease inhibitor cocktail prepared according to manufacturer's instructions (Sigma-Aldrich) and 1 mg ml⁻¹ lysozyme (Bioshop). Cells were incubated on ice for 30 min and then disrupted by sonication with a Branson 250 Sonifier with six cycles of 80% power (15 s) and intermittent cooling (30 s). The crude lysate was clarified by centrifugation, and the His-tagged fraction in the supernatant was purified by Qiagen Ni-NTA agarose (Invitrogen) using manufacturer's instructions. The suspension was washed four times with the corresponding extraction buffer containing 20 mM imidazole and eluted with buffer containing 250 mM imidazole. The purified protein fraction was desalted and concentrated using 3k MWCO Amicon Ultra-15 centrifugal filter units (Sigma-Aldrich). Proteins were used immediately in assays or brought to 15% glycerol and stored at −80 °C. Protein concentrations were determined using the Bradford method, analyzed by SDS-PAGE, and stained with Coomassie Blue according to manufacturer's instructions (Bio-Rad).

Enzyme assays were carried out in a reaction volume of 100 μl. Dehydratase (EDD and DHAD) assay buffer consisted of 100 mM Tris-HCl (pH 8.0) with 10 mM $MgCl_2$ and 1 mM dithiothreitol (DTT) and substrate (6-PG or DIV) at 4.4 mM. Aldolase enzymes assays were conducted in 50 mM $KH_2PO_4$ (pH 8.0), 100 mM NaCl, 1 mM DTT and substrate (KDPG, fructose-1,6-bisphosphate (FBP), or dihydroxyacetone phosphate (DHAP) and erythrose 4-phosphate (E4P)) in a concentration range of 0.25–1 mM. Following addition of 20 μg purified recombinant enzyme, enzyme assays were incubated for 1 h at 30 °C and quenched by addition of 1 vol of chloroform, vortexed, and incubated on ice for 10 min. Coupled assays were performed in dehydratase assay buffer with 2.9 mM 6-PG. Reaction mixtures were centrifuged, and 50 μl of the upper phase was transferred to a vial containing 50 μl of acetonitrile for analysis by LCMS/MS. An aliquot of the aldolase reaction with KDPG was derivatized for pyruvate analysis by GC-MS as described below. Control reactions lacking substrates or conducted with boiled protein were included in parallel.

### Enzyme assay from total plant protein

All protein extraction steps were done at 4 °C. Soybean leaf tissue (5 g) was ground in liquid nitrogen and extracted in a blender containing 2.5 g polyvinylpolypyrrolidone (PVPP) and 100 ml extraction buffer (100 mM Tris-HCl pH 8.0, 20 mM $MgCl_2$, 10% glycerol, 0.05% tween-20, 2 mM DTT, 1% polyvinylpyrrolidone, 1% protease inhibitor cocktail). The extract was then centrifuged at $4000 \times g$ for 45 min and clarified by filtration through Whatman filter paper. The crude extract was exchanged into assay buffer (100 mM Tris, 100 mM KCl (pH 8.0 and 10% glycerol) using a Bio-rad Econo-Pac® 10DG desalting column.

### Confocal microscopy

Subcellular localization of proteins was established by *Agrobacterium tumefaciens* infiltration of *N. benthamiana* using the pB7FWG2 construct[55] that encoded the full-length recombinant GmEDA protein fused to eGFP at its C-terminus. *Agrobacterium tumefaciens* strain GV3101 was transformed with 1 μg purified plasmid and selected on LB

plates containing rifampicin (50 μg ml⁻¹), gentamycin (50 μg ml⁻¹), and spectinomycin (50 μg ml⁻¹). Single colonies were grown in a 5 ml starter culture at 28 °C for 2 days containing the same antibiotics. This culture was centrifuged at $1500 \times g$ for 30 min, washed in infiltration medium (10 mM MES pH 5.7 with 10 mM $MgCl_2$), pelleted a second time, and resuspended in infiltration medium containing 200 μM acetosyringone to an $OD_{600}$ of 0.8. Each culture was mixed with an equal volume of a culture expressing the HCPro silencing suppressor protein[56]. Cultures were infiltrated into 2–4 *N. benthamiana* leaves with a plastic syringe and returned to the growth chamber for 2 days. Leaf sections (-0.5 cm²) were cut and mounted on slides in 5 mM MES pH 6.0 with 10% glycerol. Images were captured on a Zeiss Axio Observer Z1 LSM 800 confocal microscope with the Plan-Apochromat 40x/1.3 Oil DIC (UV) VIS-IR M27 objective lens. eGFP excitation was done at 488 nm with a capture window of 500–520 nm, whereas chlorophyll autofluorescence was observed by excitation at 561 nm and emission up to 750 nm. Images were generated with a resolution of 2048 × 2048 pixels (16 bits per pixel). Z-stacks separated by 26.95 μm were merged for each channel (eGFP, chlorophyll, brightfield) based on the average pixel intensity using ImageJ with identical contrast and brightness settings applied to each sample.

### Extraction of plant and bacterial tissues for KDPG detection

All steps of sample harvest, extraction, and data acquisition comply with recommendations for metabolomics annotation, quantification, and reporting practices[57], as described below. Plant and bacterial tissue was flash frozen in liquid nitrogen, ground to a fine powder in a mortar and pestle, and lyophilized to dryness against a vacuum of 25 μbar. Freeze dried tissue was stored in a sealed container at −20 °C until extracted. All extraction steps were performed at 4 °C. To detect KDPG in biological tissue, lyophilized Arabidopsis, soybean, *Solanum lycopersicum* (tomato), *N. benthamiana, C. vulgaris*, and *Chlamydomonas reinhardtii* tissues were extracted at 4 °C with 4 ml ice-cold extraction buffer containing 50% (*v/v*) acetonitrile containing 5 mM ammonium acetate (pH 9.0) using a procedure based on Evans et al.[58]. After vortexing for 1 h, the extract was centrifuged for 30 min at $4000 \times g$. The supernatant was lyophilized to dryness overnight, resuspended in 200 μl 10 mM ammonium acetate, and back extracted with 1 vol $CHCl_3$. The upper, aqueous phase was mixed with 1 vol acetonitrile and filtered (0.2 μm polytetrafluoroethylene) and analyzed within 24 h of preparation. Plant extractions were performed with 100–500 mg of tissue to establish the optimal ratio of plant tissue to solvent and minimize ion suppression. For microbial cultures, 200 ml liquid culture were pelleted by centrifugation following overnight growth at 28 °C (*A. tumefaciens*) or 37 °C (*E. coli*). *Saccharomyces cerevisiae* and *S. pombe* were cultivated in YES media at 32 °C until an $OD_{600nm}$ of 0.8 was reached. Bacteria and yeast cells were resuspended in 5 ml of ice-cold acetonitrile/methanol/water (2:2:1 v/v/v) with 0.1% ammonium hydroxide using a procedure adapted from Gonzalez-Cabanelas et al.[59]. The suspension was incubated on ice for 15 min, sonicated, and centrifuged at $4000 \times g$ for 40 min. The supernatant was lyophilized to dryness, resuspended in 500 μl aqueous 10 mM ammonium acetate pH 9.0, and back extracted with 1 vol of chloroform. The upper aqueous phase was diluted with 1 vol of acetonitrile, filtered, and analyzed by LCMS/MS.

Additional measures taken for quality assurance at the sample extraction phase are as follows. A control tissue sample consisting of ground, lyophilized *A. thaliana* tissue was extracted with each batch as a QC sample for batch correction. An internal standard (2-deoxyglucose-6-phosphate; DGP) was added to each sample (12 ng mg⁻¹ DW) to calculate recovery by comparison to no tissue controls with the same amount of internal standard. Quantification of KDPG detected in microbial extracts was calculated with an external calibration standard constructed from authentic standards containing DGP. For samples where KDPG was detected, quantification was repeated using standard

addition by adding KDPG at the beginning of the extraction ($n = 3$ technical replicates). Sample order was randomized, except for calibration samples which were run at the beginning and end of each sequence to confirm consistent retention times.

## Metabolite analysis

LCMS/MS analyses were carried out using an Agilent series II 1290 UHPLC coupled to a Sciex 4500Qtrap triple quadrupole mass spectrometer operating in negative ionization mode. For all phosphory-lated samples, chromatographic separation was accomplished by hydrophilic interaction chromatography using an XBridge BEH amide column (2.1 mm × 150 mm; Waters Corporation) fitted with a guard column containing the same sorbent (2.1 mm × 5 mm) with flow at 0.5 ml min⁻¹. Liquid chromatography conditions started with 0% solvent A (20 mM ammonium bicarbonate pH 10.5) and 100% B (20 mM ammonium bicarbonate pH 10.5 in 80% acetonitrile), reaching 16% A by 5 min and holding until 10 min. A steep gradient occurred from 10–11 min reaching 40%, which was held for 4 additional min. This was followed by a step change to initial conditions for 15 min (30 min total). Branched chain intermediates were analyzed on a Luna C-18(2) column (100 mm × 2.0 mm, 2.5 μm particle size; Phenomenex) whose buffers consisted of 5 mM tributylamine with 15 mM acetic acid (pH 4.9) (A) and acetonitrile (B). The gradient was as follows: 0–45% B (0–12 min), 45–60% B (12–14 min), 60–90% B (14–15 min), and 0% B (15–20 min) at a flowrate of 0.25 ml min⁻¹. Sample injection volume was 5 μl, and the column temperature was maintained constant at 25 °C. Extracts were kept in a chilled autosampler and analyzed within 24 h of preparation. Authentic standards, obtained from Millipore-Sigma unless otherwise noted, were used to optimize detection in bacteria and plant extracts and enzyme assays. Sciex Analyst software (v1.7.2) was used for data acquisition, and SciexOS 2.0 was used for all data analysis.

Multiple reaction monitoring of analytes was as follows: KDPG ($m/z$ 257 → 79; DP −20 V, EP −10 V, CE −61 V, CXP −5 V); 6-PG ($m/z$ 275 → 97; DP −35 V, EP −10 V, CE −62 V, CXP −7 V); GAP ($m/z$ 169 → 97; DP −35 V, EP −10 V, CE −20 V, CXP −20 V); 2-DGP ($m/z$ 243 → 79; DP −35 V, EP −10 V, CE −62 V, CXP −7 V), sedoheptulose-1,7-bisphosphate ($m/z$ 369 → 79; DP −50 V, EP −10 V, CE −76 V, CXP −7 V). MS/MS product ion spectra are shown in Supplementary Fig. 6. BCAA intermediates were monitored in negative ionization mode using Q1 multiple ion scans with the following settings: DIV ($m/z$ 133; DP −65 V, EP −10 V); KIV ($m/z$ 115; DP −147, EP −10 V).

Pyruvate was analyzed by gas chromatography–mass spectrometry (GCMS) based on Bergman et al.[60], which consisted of evaporating the enzyme assay to dryness under nitrogen gas, followed by resuspension in 100 μl pyridine containing 20 mg ml⁻¹ methoxylamine and incubation at 30 °C for 90 min. A 30 μl aliquot was transferred to a glass vial, combined with an equal volume of N-methyl-N-(trimethylsilyl)trifluoroacetamide, and sealed under a nitrogen atmosphere. One μl was analyzed by split injection (1:10) on an Agilent 7890B GC system fitted with a VF-5ms capillary column (Agilent Technologies) and a 5977B mass selective detector operating in electron impact ionization mode (50 eV) set to scan from $m/z$ 50–550. Helium carrier gas flow was set to 1.1 ml min⁻¹, and the oven temperature was isothermal for 1 min at 70 °C, followed by an increase to 325 °C with a final hold time of 5 min. Agilent MassHunter 10.0 was used for all GCMS data acquisition and analysis.

All mass spectrometry detection parameters were uploaded to MetaboLights (https://www.ebi.ac.uk/metabolights/) under accession MTBLS9012, and all chromatographic and spectral data are available in the Source Data file accompanying this article.

## Phylogenetic analysis of EDD sequences

To find orthologs of EDD in public databases, the *E. coli* EDD sequence (ACX39449.1) was used as a query for the NCBI BlastP algorithm (May 2023 version) with an e-value cutoff of $10^{-10}$.

Putative *EDD* and *DHAD* sequences from bacteria, archaea, cyanobacteria, fungi, algae and plants were downloaded from NCBI GenBank. Sequences from members of the Zygnematophyceae were retrieved by tBLASTn search on NCBI Transcriptome Shotgun Assembly (TSA) database (see Supplementary Table 1). A total of 155 sequences were aligned in MEGA X and employed for phylogenetic analysis. The multiple sequence alignment was generated using the MUSCLE algorithm in MEGA X[61], and a phylogram was constructed using the maximum likelihood (ML) and the Jones-Taylor-Thornton substitution model with 1000 bootstrap replicates to evaluate tree topology.

## Protein structural modeling and substrate docking

The structure of Arabidopsis DHAD in complex with an iron sulfur cluster (PDB 5ZE4) and an AlphaFold model[26] generated from the *E. coli* K12 EDD sequence (Uniprot ID P0ADF6) were used in substrate docking simulations. The predicted N-terminal transit peptide from the DHAD structure was removed before docking analysis. The dimeric dehydratase structures were generated using crystallographic symmetry from PDB 5ZE4. The iron-sulfur cluster and magnesium ion coordinates were modeled in the EDD structure using those of PDB 5ZE4. Molecular docking was performed using AMDock's graphical user interface for AutoDock 4.2.6[62,63]. The .mol2 files of ligands were downloaded from the PDB and used without modification. All inputs were converted to pdbqt format within AutoDock4 using AutoDockTools. Visualization, structure analysis, and figure generation were performed in ChimeraX[64].

## Reporting summary

Further information on research design is available in the Nature Portfolio Reporting Summary linked to this article.

# Data availability

The data supporting Fig. 1 are available in Supplementary Table 1. The data supporting Supplementary Fig. 2 were downloaded at doi: 10.6084/m9.figshare.7629383. All raw mass spectrometry data are available under accession MTBLS9012 at the MetaboLights data repository (https://www.ebi.ac.uk/metabolights/search). Source data are provided with this paper.

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

## Acknowledgements

This study was funded by a Discovery grant from the Natural Sciences and Engineering and Research Council (NSERC) of Canada (RGPIN-2017-06400 to M.A.P.; RGPIN-2023-05615 to M.A.C.) and a John Evans Leadership Fund grant from the Canadian Foundation for Innovation (36131 to M.A.P.; 40684 to M.A.C.). The authors also acknowledge a generous NSERC CGSD graduate scholarship supporting M.E.B. and an NSERC CGSM scholarship supporting A.F. The authors thank the staff of the UTM Teaching Greenhouse and Academic Machine Shop for their excellent technical support. The authors finally thank Prof Saša Stefanović for stimulating discussion.

## Author contributions

S.E.E. and A.E.F. performed main experiments. M.E.B. performed supporting experiments and assisted in data analysis. N.S.S. and M.A.C. carried out protein structural modeling. M.A.C. and M.A.P. wrote the manuscript. M.A.P. directed the research.

## Competing interests

The authors declare no competing interests.
