## [Peer Review File · Nature Communications]

Plastid ancestors lacked a complete Entner-Doudoroff pathway, limiting plants to glycolysis and the pentose phosphate pathwayReviewer #1 (Remarks to the Author):

This manuscript reports the absence of a fully functional ED pathway in plants and potentially their plastid ancestor. To support this claim, authors showed the absence of EDD enzyme in plants through phylogenetics and conducted substrate docking to confirm the substrate specificity for EDD and DHAD, two structurally similar dehydratase. The *in vitro* biochemical assays for EDD activities using either genuine EDDs or structurally similar DHAD candidates further validated the substrate docking simulation. In addition, the metabolite analysis of KDPG indicates its absence in many eukaryotes. This result is in contradiction with a previous report that KDPG could be detected in at least barley roots. Overall, I found the evidence convincing to illustrate the absence of the EDD gene in the plant domain, thus supporting the notion that the ED pathway is absent in plants.

The prevalence of EDA enzyme in all life forms thus is intriguing to understand its potential contribution to cell metabolism. The authors further showed that EDA has moonlighting aldol cleavage activities toward several phosphate sugars. The confocal microscopy is also a nice touch to show the subcellular localization of EDA in plants.

In summary, I believe the main result here is significant in that it provides a right path for future studies to understand the potential role of EDA in contributing cell metabolism, rather than through the ED pathway activity. I also like some of the discussion authors provided in terms of how ED activity could divert fluxes out of CBB or cancel G6P shunt fluxes if it was to exist in either chloroplast or cytosol. The method description is also thorough in describing different procedures. I have some additional minor comments listed below:

- The previous research in cyanobacteria labeled DHAD as a potential candidate for EDD activities. In the substrate docking simulation, did authors try docking 6PG into DHAD for potential interactions or did it fail to dock?
- Please correct the mislabeling in the Figure 3 legends. Fig. 3A, B, C should be B, C, D.
- Please provide the MS/MS spectrum of 6PG, KDPG, and GAP either in the main figures or in supplemental figures to support Figure 3 and Figure 4 conclusions.
- Line 52-53: rephrase "and the availability of non-glycolytic sources of ATP..."
- Line 100: I think authors meant "in vitro" rather than "in vivo" for plant EDA activity in Ref 11?
- Line 312: Fig 2C is DHAD substrate domain. Here should be Fig 2B to reference *E. coli* EDD domain 1. Again, Fig. 2D should be 2B as well in line 315.
- Line 319-: "promoter"? Do you mean monomer?
- Line 324: here should be Fig. 2C and 2D
- Line 461-462: Please double check the accuracy of this statement regarding non-phosphorylating ED in *Synechocystis*.

Reviewer #2 (Remarks to the Author):

In this interesting manuscript, the authors use different methodologies to shed light on the evolutionary history and distribution of the Entner-Doudoroff (ED) pathway.

The manuscript text and figures are very clear. I am very pleased by the complementary bioinformatics and experimental approaches used by the authors, which I find adequate and

elegant.

The general conclusions of the study are convincing and important. The results support the notion that the ED pathway is absent from eukaryotic lineages and allow proposing that the plastid ancestor likely lacked 6-PG dehydratase (EDD), and hence a complete ED pathway.

The authors also discuss relevant points, including an alternative hypothesis for the lack of EDD in photosynthetic eukaryotes, the persistence of KDPG aldolase (EDA) in plants, and the potential exploitation of the ED pathway in plant synthetic biology approaches.

I recommend publication.

Minor comments:

1. It would have been interesting to explore the physiological and metabolic impact of localizing EDD to the plant cytosol, plastids, and mitochondria.
2. Line 353: There is a typo; please change assaus to assays.
3. Line 361: Perhaps "not detected" is better than "blocked" in this context.
4. Figure 1C: "Bacteria" could be centered like the other biological lineages names. The curved blue line could be separated from "Cyanobacteria".
5. Figure 3A: Retention times 12 and 4 for 6-PG, KDPG, and GAP should be separated.
6. Figure 4: There is a blue line on the left side of the figure that goes from the KDPG standard to the Sc+KDPG.
7. Figure 5E: I suggest including zoomed images of the confocal microscopies.

Reviewer #3 (Remarks to the Author):

In their article "Plastid ancestors lacked a complete Entner-Doudoroff pathway, limiting plants to glycolysis and the pentose phosphate pathway", Evans et al. describe a well-conducted study into a big question in the evolution of plant biochemistry: the distribution of a functioning ED pathway. Overall, the study is of good quality. I however have remarks regarding the interpretation / framing.

Regarding data availability: have the raw spectra data from the mass spec been uploaded? E.g., to MetaboLights or another adequate database?

I think that the study is of great value and the analyses carried out are an important contribution to the scientific discourse. Yet, I am not convinced by the evidence for the "plastid ancestor" (highlighted in the title and throughout). As far as I can judge, the entire statement rests on the following: "Although the true sister group to plastids is uncertain, proposed candidates for the most recent common ancestor to plastids and free-living cyanobacteria coalesce around *Gleomargarita lithospora* and several closely related taxa which include *Synechococcus* sp., *Synechocystis* sp., *Prochlorococcus marinus*, *Trichodesmium* sp., *Oscillatoria* sp., and *Arthrospira* sp."

When we consider features of the cyanobacterial progenitor of plastids, we are talking about an organism that likely lived around 2 billion years ago. Screening a few cyanobacteria is not enough. Prokaryotes are known to lose and gain genes with ease, the latter propelled by LGT. The problems that result from this are expertly explained in: Endosymbiotic gene transfer from prokaryotic pangenomes: Inherited chimerism in eukaryotes by Ku C, Nelson-Sathi S, ..., and Martin WF. PNAS 2015 112 (33) 10139-10146 <https://doi.org/10.1073/pnas.1421385112> When talking about these extant organisms (leaves on the tree from this year and not the 2-billion-year-old phylogenetic entity that would branch off a deep node on the tree), please do not

talk about them them as if they would represent 1to1 the cyanobacterial plastid progenitor (which lived about 2 billion years ago!). Talk about the groups that they belong to. Talk about the last common ancestors that they might share with the cyanobacterial plastid progenitor. Talk in trees.

"none of which is considered a candidate for the ancestor of plastids." -> Again, these are all extant organisms and not an organism that lived 2 billion years ago. See above. And when it comes to the groups of filamentous nitrogen fixing cyanos (like Nostoc), this is also not true. See, e.g., Dagan et al 2013 GBE -- DOI: 10.1093/gbe/evs117 -- for a discussion on this shifting target, see: <https://doi.org/10.1016/j.cub.2016.12.006>

Figure 1: It would help a lot to reconcile the gene phylogeny here with a species tree to show the distribution. Also the choice of organisms is unclear. A lot of Chlorella/Chlorella-like algae have been picked, but only one streptophyte alga, not including the closest algal relatives to land plants, see, cite and add: <https://www.biorxiv.org/content/10.1101/2023.01.31.526407v1> & <https://www.nature.com/articles/s41477-023-01491-0#Abs1> & <https://www.sciencedirect.com/science/article/pii/S0092867420304827> & <https://doi.org/10.1016/j.cub.2022.08.022>

Furthermore, please cite all the genome data that have been used--I do not find them in the reference list or material and methods e.g. the recently sequenced Chlorelloids or the key species *Chara braunii*.

The biochemical data and figures are, as far as I can judge, all sound and of adequate quality. See comment about data availability above. The only aspect that I would like the authors to consider here is to also provide in the figures 3 to 5 the phylogenetic context from which the enzymes were derived would be very helpful (writing e.g. embryophytes, Chlorophyceae etc. below the species abbreviation would help already a lot)

Responses to reviewers letter

We humbly thank the reviewers of our manuscript for dedicating their time to this review. Below we provide an itemized list of actions taken in response to their observations (our responses appear in blue text).

Reviewer #1

- The previous research in cyanobacteria labeled DHAD as a potential candidate for EDD activities. In the substrate docking simulation, did authors try docking 6PG into DHAD for potential interactions or did it fail to dock?

Yes, we did simulate docking of 6PG into DHAD as well as DIV into EDD. Neither substrate could be docked into the reciprocal active sites without exceeding our significance threshold (Autodoc). We changed the caption in figure 2 and the results text describing these docking experiments to better highlight these results, which support highly specific dehydratases with no overlap in substrate specificity.

- Please correct the mislabeling in the Figure 3 legends. Fig. 3A, B, C should be B, C, D.

Done

- Please provide the MS/MS spectrum of 6PG, KDPG, and GAP either in the main figures or in supplemental figures to support Figure 3 and Figure 4 conclusions.

Done. We have created a new supplemental figure (S1) showing MS/MS spectra of 6PG, KDPG, and GAP, as requested. A line was added to the methods to reference this new figure, and the other supplemental figures were renumbered accordingly to reflect their order of appearance.

- Line 52-53: rephrase “and the availability of non-glycolytic sources of ATP...”

Done. This sentence has been rewritten as “Organisms which can produce ATP through photosynthesis or aerobic respiration are less subject to this constraint and may favor the ED pathway..”

- Line 100: I think authors meant “in vitro” rather than “in vivo” for plant EDA activity in Ref 11?

Fixed, thank you

- Line 312: Fig 2C is DHAD substrate domain. Here should be Fig 2B to reference E. coli EDD domain 1. Again, Fig. 2D should be 2B as well in line 315.

Thank you for pointing this out. All 3 corrections have been made. The caption was further rewritten for clarity.

- Line 319-: “promoter”? Do you mean monomer?

We have used the term “protomer” rather than “monomer” here to emphasize the conformational changes of the monomer into the shape it takes in the functional, active dimer.

- Line 324: here should be Fig. 2C and 2D

Fixed, thank you.

- Line 461-462: Please double check the accuracy of this statement regarding non-phosphorylating ED in *Synechocystis*.

In the citation in question, Chen et al (2016, Proc Natl Acad Sci) speculate that the *Synechocystis* DHAD protein *slr0452* might participate in the semi-phosphorylating ED pathway “We hypothesize that the encoded enzyme might be involved in the synthesis of the branched chain amino acids valine and isoleucine [sic], as well as the conversion of gluconate to KDPG...” They also showed *Synechocystis* EDA protein cleaves KDPG but not KDG: “This establishes that *Sll0107* does not operate in a nonphosphorylating ED pathway” In the semi-P pathway, KDG is phosphorylated by a kinase, which has only been shown in a thermophilic archeon species *Sulfolobus*. We have amended this statement and added two references (Lambley et al 2004, Kourli 2013) to reflect the literature more accurately and better represent the argument posed by Chen et al.

Reviewer #2:

1. It would have been interesting to explore the physiological and metabolic impact of localizing EDD to the plant cytosol, plastids, and mitochondria.

We have briefly mentioned this topic in the discussion, but we feel it is premature to speculate too broadly on the potential impact of adding the ED pathway to these compartments.

2. Line 353: There is a typo; please change assays to assays.

Fixed, thank you

3. Line 361: Perhaps “not detected” is better than “blocked” in this context.

Changed as requested

4. Figure 1C: “Bacteria” could be centered like the other biological lineages names. The curved blue line could be separated from “Cyanobacteria”.

The cited issues with fig 1C have been addressed as requested. The label “bacteria” was in fact centered for bacterial DHAD sequences, but the figure did not make the EDD/DHAD border clear. We have emphasized the border between EDD and DHAD sequences and re-colored the branches of the dendrogram with the same blue color scheme for clarity. We have also added number of new sequences.

5. Figure 3A: Retention times 12 and 4 for 6-PG, KDPG, and GAP should be separated.

Done

6. Figure 4: There is a blue line on the left side of the figure that goes from the KDPG standard to the Sc+KDPG.

Fixed, thank you.

7. Figure 5E: I suggest including zoomed images of the confocal microscopies.

We have included a new zoomed in image showing colocalization of EDA with chloroplasts (figure 5F). Our EDA-eGFP fusion produces small speckles of fluorescence which are possibly aggregates, but it does localize with chloroplasts in *N. benthamiana*. We have added two new citations which further document this fluorescence pattern of chloroplast localized, GFP tagged proteins (Di and Rodriguez-Concepcion, *Plants* 2023 and Krieger et al *IJMS* 2021).

Reviewer #3

Regarding data availability: have the raw spectra data from the mass spec been uploaded? E.g., to MetaboLights or another adequate database?

Submission of our raw spectral data to MetaboLights is in progress. The corresponding link will be added to our manuscript as soon as it is finalized. In this revised version, we have also included a source data file which includes most of the same information.

I think that the study is of great value and the analyses carried out are an important contribution to the scientific discourse. Yet, I am not convinced by the evidence for the "plastid ancestor" (highlighted in the title and throughout). As far as I can judge, the entire statements rests on the following: "Although the true sister group to plastids is uncertain, proposed candidates for the most recent common ancestor to plastids and free-living cyanobacteria coalesce around *Gleomargarita lithospora* and several closely related taxa which include *Synechococcus* sp., *Synechocystis* sp., *Prochlorococcus marinus*, *Trichodesmium* sp., *Oscillatoria* sp., and *Arthrospira* sp."

When we consider features of the cyanobacterial progenitor of plastids, we are talking about an organism that likely lived around 2 billion years ago. Screening a few cyanobacteria is not enough. Prokaryotes are known to lose and gain genes with ease, the latter propelled by LGT. The problems that result from this are expertly explained in: Endosymbiotic gene transfer from prokaryotic pangenomes: Inherited chimerism in eukaryotes by Ku C, Nelson-Sathi S, ..., and Martin WF. *PNAS* 2015 112 (33) 10139-10146 <https://doi.org/10.1073/pnas.1421385112>

When talking about these extant organisms (leaves on the tree from this year and not the 2-billion-year-old phylogenetic entity that would branch off a deep node on the tree), please do not talk about them as if they would represent 1 to 1 the cyanobacterial plastid progenitor (which lived about 2 billion years ago!). Talk about the groups that they belong to. Talk about the last common ancestors that they might share with the cyanobacterial plastid progenitor. Talk in trees.

We thank reviewer 3 for these comments and have endeavored to use them to strengthen our manuscript. We have revised the presentation and discussion of the phylogenetic evidence and included the citations requested by reviewer 3. We feel our revised version can now answer the following question: do plants lack the ED pathway because the ancestors of plastids were not ED pathway-utilizing cyanobacteria or were these genes lost through deletion during endosymbiosis? In this updated version, we have strengthened our evidence that photosynthetic eukaryotes don't use the ED pathway because the plastid ancestor lacked an EDD gene.

These revisions can be found in the Results section beginning with "About a quarter of heterotrophic bacteria use the ED pathway..." and in the Discussion beginning with "Plastids are nested within cyanobacterial lineages..."

"none of which is considered a candidate for the ancestor of plastids." -> Again, these are all extant organisms and not an organism that lived 2 billion years ago. See above. And when it comes to the groups of filamentous nitrogen fixing cyanos (like Nostoc), this is also not true. See, e.g., Dagan et al 2013 GBE -- DOI: 10.1093/gbe/evs117 -- for a discussion on this shifting target, see: <https://doi.org/10.1016/j.cub.2016.12.006>

We have expanded and updated our explanation accordingly, as suggested by reviewer 1. We have read and cited the recommended papers and incorporated them into our discussion and description of results. This includes an updated discussion of plastid sister groups, their common ancestors, and how the current state of phylogenomic analysis of plastid origins can illuminate the present question of the distribution of the ED pathway.

Figure 1: It would help a lot to reconcile the gene phylogeny here with a species tree to show the distribution. Also the choice of organisms is unclear. A lot of Chlorella/Chlorella-like algae have been picked, but only one streptophyte alga, not including the closest algal relatives to land plants, see, cite and add: <https://www.biorxiv.org/content/10.1101/2023.01.31.526407v1> & <https://www.nature.com/articles/s41477-023-01491-0#Abs1> & <https://www.sciencedirect.com/science/article/pii/S0092867420304827> & <https://doi.org/10.1016/j.cub.2022.08.022>

We have expanded our single gene phylogeny with members of the Zygnematophyceae and other species as requested and updated Fig 1C. We have also included a species phylogeny of cyanobacteria as requested (see supplemental figure S7) to help explain the distribution of EDD genes among crown cyanobacteria, which we argue is the result of lateral gene transfer rather than vertical inheritance. Several major advances in plastid/cyanobacterial phylogeny have appeared in the literature in recent years using sophisticated bioinformatic analyses (Moore et al 2019, Ponce-Toledo 2017, Ochoa de Alda 2014). It is beyond the scope of our study to attempt to improve upon these, and we have therefore reconstructed a species tree using branch length data obtained from the most recent of these studies. Using this species tree to interpret our EDD gene phylogeny, we show that EDD occurrence among cyanobacteria is rare and sporadic. Its distribution among cyanobacteria is consistent with its absence in the last common ancestor between cyanobacteria and plastid-bearing eukaryotes. We therefore reject the alternative explanation that evolutionary forces compelled its deletion. Interestingly, this implies there is still a chance that plants can be rewired to use this alternative pathway, which we note in our revised discussion.

Fig 1C is not intended to exhaustively survey the distribution of DHAD genes, but rather to illustrate our general conclusion: that the DHAD gene tree clusters according to kingdom and reflects the commonly accepted phylogeny of these groups, while EDDs clustering is intermixed and shows evidence of lateral gene transfer. We have also modified the text describing this figure to better define its purpose, justify the choices of sequences we included, and clarify the conclusions we have drawn here to the inferred distribution of the ED pathway among extant organisms. Here we have added citations to Hess et al (2022), Jiao et al (2023), Feng et al (2023), and Dadras et al (2023), as suggested, and referenced the new supplemental figure to illustrate our conclusions. We hope that with these changes, the point of figure 1C is now clear.

Furthermore, please cite all the genome data that have been used--I do not find them in the reference list or material and methods e.g. the recently sequenced *Chlorelloids* or the key species *Chara braunii*.

All sequences were obtained from NCBI May 2023. We have added a note to this effect in the methods section. In this version, we have added several genomic resources, and references to those studies have now been added to the methods section or the corresponding supplemental figure (see supplemental table S2 for instance).

The biochemical data and figures are, as far as I can judge, all sound and of adequate quality. See comment about data availability above. The only aspect that I would like the authors to consider here is to also provide in the figures 3 to 5 the phylogenetic context from which the enzymes were derived would be very helpful (writing e.g. embryophytes, Chlorophyceae etc. below the species abbreviation would help already a lot)

Thank you for pointing this out. Of the figures mentioned (3-5), figure 4 has the most phylogenetic context worth highlighting. We have added phylogenetic information as requested (vertical colored bars) and re-colored the chromatograms and labels to match those of figure 1C for consistency. In figure 3, we are contrasting the biochemical activity of an EDD (which necessarily comes from a bacterial source) to typical DHADs from plant and bacterial sources (which we show to be functionally equivalent). The only important distinction here in terms of biological source is plants vs bacteria. We have re-colored the chromatograms to use the same color scheme in other figures and more clearly pointed out this distinction in the revised caption. For figure 5, this is also true. We have simply selected typical EDA sequences from plants and bacteria and shown them to be biochemically equivalent and have similar kinetic properties. We hope that these modifications to fig 4 to highlight the phylogenetic coverage of our KDPG analysis, along with minor changes to figs 3 and 5, have addressed reviewer concerns on this point.

Reviewer #1 (Remarks to the Author):

The authors have addressed my comments. Two minor things for authors to consider:

- "The domain 1 consensus sequence that we observed exclusively in EDD enzymes encompasses $\alpha 3$, the first four amino acids of $\alpha 4$, and the connecting $\alpha 3$ - $\alpha 4$ loop in E. coli EDD (figure 2C)" Here is Fig 2B not 2C. I suggest authors carefully examine their manuscript to make sure all the details are correct.
- Since EDA has alternative aldolase activity toward FBP and SBP, I suggest authors provide full kinetics data k_{cat}/K_m rather than just K_m . It could provide more insight on its potential role in the Calvin cycle.

Reviewer #2 (Remarks to the Author):

The authors have addressed all my suggestions and questions. I congratulate the authors on this exciting study.

Reviewer #3 (Remarks to the Author):

The authors have fully and satisfactorily addressed all my remarks. I do not have any further comments and congratulate the authors on a very interesting paper.

Response to reviewers

Reviewer #1

The authors have addressed my comments. Two minor things for authors to consider:

- “The domain 1 consensus sequence that we observed exclusively in EDD enzymes encompasses $\alpha 3$, the first four amino acids of $\alpha 4$, and the connecting $\alpha 3$ - $\alpha 4$ loop in *E. coli* EDD (figure 2C)” Here is Fig 2B not 2C. I suggest authors carefully examine their manuscript to make sure all the details are correct.

Error has been corrected with thanks. In the revised numbering, the text now refers to Fig. 2d (the EDD structure).

- Since EDA has alternative aldolase activity toward FBP and SBP, I suggest authors provide full kinetics data k_{cat}/K_m rather than just K_m . It could provide more insight on its potential role in the Calvin cycle.

Although we are working on this, we do not currently have this data. We intend to produce a followup paper focusing on kinetics of EDA which is beyond the scope of the present manuscript.

Reviewer #2 (Remarks to the Author):

The authors have addressed all my suggestions and questions. I congratulate the authors on this exciting study.

We thank reviewer 2 for their time and dedication to improving our manuscript.

Reviewer #3 (Remarks to the Author):

The authors have fully and satisfactorily addressed all my remarks. I do not have any further comments and congratulate the authors on a very interesting paper.

We thank reviewer 3 for their time and encouraging comments.